# How does COVID-19 affect electoral participation? evidence from the French municipal elections

Abdul Noury[1]* , Abel François[2] , Olivier Gergaud[3] , Alexandre Garel[4]

**1** Division of Social Sciences, New York University Abu Dhabi, Saadiyat Island, Abu Dhabi, United Arab Emirates, **2** LEM (UMR 9221), University of Lille, Lille, France, **3** KEDGE Business School (Bordeaux Campus), Talence, France, **4** Audencia Business School, Nantes, France

These authors contributed equally to this work.
* agn2@nyu.edu

**Data Availability Statement:** The data underlying the results presented in the study are available from Harvard Dataverse at https://dataverse.harvard.edu/dataset.xhtml?persistentId=doi:10.7910/DVN/XV1VSG.

## Abstract

This article investigates the effects of the COVID-19 outbreak on electoral participation. We study the French municipal elections that took place at the very beginning of the ongoing pandemic and held in over 9,000 municipalities on March 15, 2020. In addition to the simple note that turnout rates decreased to a historically low level, we establish a robust relationship between the depressed turnout rate and the disease. Using various estimation strategies and employing a large number of potential confounding factors, we find that the participation rate decreases with city proximity to COVID-19 clusters. Furthermore, the proximity has conditioned impacts according to the proportion of elderly –who are the most threatened– within the city. Cities with higher population density, where the risk of infection is higher, and cities where only one list ran at the election, which dramatically reduces competitiveness, experienced differentiated effects of distance.

## Introduction

The coronavirus pandemic (COVID-19) not only threatens the health of the population, and major sections of the economy, it also challenges elections throughout the democratic world. The fear of becoming infected with the virus may cause selective participation, where a non-negligible fraction of voters, particularly those with higher health risks (such as elderly and vulnerable voters), abstain from voting. Selective participation may lead to reduced legitimacy of elected representatives and open the door to controversies, and may eventually trigger social and political polarization and conflicts.

In this context, we provide empirical evidence for the consequences of COVID-19 for electoral participation. To do that, we scrutinize the 2020 French local elections whose first round was organized at the very beginning of the coronavirus pandemic. These run-off elections, held every six years to elect mayors, form a major poll that concerns all 34,968 municipalities of metropolitan France and overseas *département*s and territories. To renew 902,465 councilors, 46,112,785 French citizens were invited to cast their ballots on Sunday, March 15, 2020

**Funding:** The author(s) received no specific funding for this work.

**Competing interests:** The authors have declared that no competing interests exist.

(first round) and Sunday, March 22, 2020 (second round). About 44.66% of eligible voters turned out on March 15, while 41.67% voted in the postponed second round on June 28 in 4,816 communes (initially, the second round was planned for March 22; the decision to postpone because of the sanitary crisis was announced on March 16). In comparison, the turnout rate was 63.5% in the first round of the previous elections in 2014. The fall in participation at this important election was surprising for two reasons. First, municipal elections are the second most popular elections at the local level in France, just after the presidential election. Second, mayors, elected in the wake of the municipal elections, are the most popular elected officials in France (see for instance this online source on French mayors).

Despite the challenges posed by COVID-19, President Emmanuel Macron, in his address to the nation the day before the election, announced his decision to maintain the first round of the elections. At the same time, he recommended that elderly and vulnerable people (i.e. those who suffer from chronic diseases such as respiratory troubles, or are impaired) should stay home. A speech announcing that elections will be held, while solemnly recommending that a significant fraction of the population abstain from voting, is highly questionable.

In this paper, we show that during this major pandemic, in-person voting was characterized by substantially depressed turnout rates. More importantly, we show that the ongoing COVID-19 sanitary crisis, which in France started in early March 2020, reduced electoral participation in the first round of the municipal elections, particularly in areas close to the main COVID-19 clusters, in municipalities with a higher fraction of people at higher risk to develop severe forms of COVID-19.

## Effects of disease outbreak on calculus of voting

Since the seminal works of Downs [1] and Riker [2], a vast literature has documented that electoral participation is affected by all kinds of impediments that raise the cost borne by individual voters, which is known as the voting calculus framework. Usually, the calculus of voting is defined as the expected benefit of voting (satisfaction associated with the preferred candidate times the probability of being the decisive voter) plus the satisfaction of voting that is independent from election outcomes minus the cost of voting [3] (for a presentation of this literature, see S1 Section in S1 Appendix). Within this framework, the cost of voting has two main components. On the one hand, it is the cost borne by people to prepare their voting decisions, such as the amount of time and resources used to collect information about the candidates, their programs, and the main election issues [4, 5]. On the other hand, people also bear a cost strictly associated with the action of voting, although this is considered to be low [6, e.g.], including the time needed to go to the polling station, waiting in line, casting the ballot, etc. Many studies have highlighted various drivers of electoral turnout affecting such opportunity cost: transportation costs [7], weather on the date of election [8, e.g.], number of simultaneous ballots [9, e.g.], day of the week [10, e.g.], holiday period [11, e.g.], as well as available voting technology and voting processes [12].

Against this background, what are the expected effects of a large-scale epidemic on electoral participation? A rapidly spreading disease distorts the cost of voting in two ways. First, the cost of voting naturally increases for infected individuals suffering from severe fatigue. Therefore, they are less likely to go to the polling station and cast their ballots. The cost is also reinforced by a voter's altruism: not participating limits the spread of the disease due to the voter's absence. Second, the cost also depends on a voter's health conditions, both physiological and psychological (chronic disease, etc.). Particularly in the COVID-19 context, it was known at the time of election that voters with certain underlying medical conditions (such as serious heart conditions, weakened immune systems, obesity, sickle cell disease, etc.) were at higher

risk (see https://www.cdc.gov/coronavirus/2019-ncov/need-extra-precautions/people-with-medical-conditions.html). To avoid carrying this risk, they limit social interactions, including casting a ballot.

## Voters' attitudes and electoral participation in times of COVID-19

An emerging literature studies the impact of the COVID-19 pandemic on political behavior, including voter turnout (S2 Section in S1 Appendix for a detailed survey of the literature). For instance, [13] analyze the effect of COVID-19 on political behavior in Bavaria and report that the dominant party in this region benefited from the crisis. Using a series of survey experiment data retrieved from social media in Canada, [14] show that the crisis is positively correlated with increased support for the government. However, focusing on the case of Spain, [15] report that the COVID-19 crisis is associated with a national bias and an inflated demand for strong technocratic and authoritarian policymakers.

Likewise, using survey data from different Western European countries, [16] compare political attitudes of respondents before and after a national lockdown. They find that the lockdown experience increased support for current decision-makers, institutions and regimes. By contrast, they do not find any effect of the lockdown on ideology or political interest, but a small positive effect on declared turnout rates.

Our study contrasts with previous ones, as we focus on objective real-world data covering millions of voters to study the impact of the COVID-19 pandemic on electoral participation. We study French municipal-level data for two different municipal elections, contrasting 2020 with 2014. We focus on municipalities with more than 1,000 inhabitants located in metropolitan France as the voting system differs significantly below this threshold. Paris, Marseille, and Lyon, as well as Corsica and overseas territories, are dropped from the study for similar reasons. Our final sample consists of 9,304 municipalities (see S4 and S5 Sections in S1 Appendix further details on the 2020 French municipal elections and the rationales for these choices).

In France, the vast number of polling stations (a maximum of 1,000 registered voters per polling station) could limit the perceived risk of being infected while voting, whereas the risk could be higher in some US states, for example, where long lines are often observed in front of polling stations. In addition, there is no other ballot held at the same time, which limits voters' waiting time compared to other elections with multiple ballots. However, proxy voting in France is very complicated, as people need to go to see a police or justice officer, which takes time and therefore increases the cost of voting, and there are no systems for early voting, voting by mail or online voting.

At a first glance, we observe that COVID-19 depressed turnout by a substantial amount—turnout is defined as the percentage of registered voters who cast a ballot, including invalid and blank ballots, in the municipality. Indeed, participation in the first round of this election was almost 20 percentage points lower than participation in 2014 (44.7% versus 63.5%). A turnout rate of 44.7% is among the lowest ever experienced in France under the Fifth Republic at a general election, and the worst if we ignore European elections and referenda. Fig 1, based on turnout data from municipalities with more than 1,000 inhabitants, illustrates this general negative trend and shows that the variance of turnout rates also increases in 2020 compared to 2014. Although several factors are likely to be behind such historically low turnout rates, COVID-19 played a leading role. According to surveys carried out on March 13 and 14, the most frequently mentioned driver for abstaining in 2020 was: "No commute because of COVID-19" (39%), ahead of other reasons such as "This election will have no impact on my everyday life" (33%), or "My ballot will not change the outcome of the election" (27%) (see https://www.ipsos.com/fr-fr/municipales-2020).

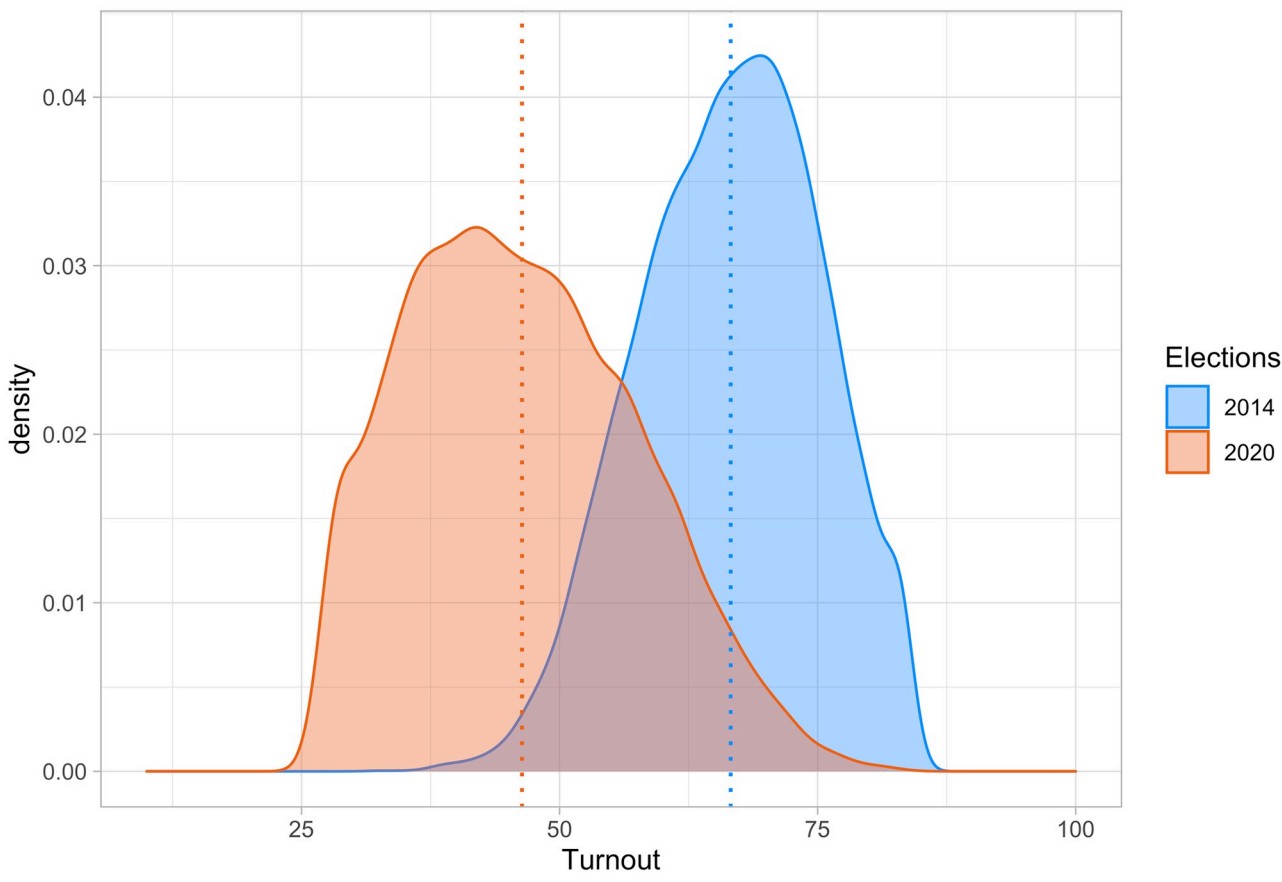

**Fig 1. Distribution of turnout in the 2014 and 2020 French municipal elections.**

The 2020 French elections took place at the beginning of the outbreak in France (see S3 Section in S1 Appendix), a period during which there was little reliable information on the virus – and only a tiny portion of the population was already sick, experiencing a higher cost of voting. As a result, the COVID-19 effect we measure corresponds to an impact on turnout through the expected cost of becoming infected by voting and the subjective probability of being infected by voting. It is clear that identifying the overall impact of COVID-19 on electoral participation is difficult, if not impossible, as we do not have a credible counterfactual for France in 2020. However, following our line of reasoning and these first observations, we hypothesize that turnout rates decrease with a higher perceived risk of being infected and higher cost of infection. The challenge is to proxy these two elements of the calculus of voting with aggregate data.

## Determinants of turnout in times of COVID-19

Since we use aggregate data (although at the municipality level, which is the finest aggregation level in France), we measure the cost and likelihood of being infected as follows:

First, to gauge the perceived risk of being infected, we focus on the seven COVID-19 clusters in metropolitan France mentioned by popular media as of March 15 (figures in parentheses indicate the zip code of the city *département*): Auray (56), Biéville-Beuville (14), Bruz (35), Crépy-en-Valois (60), La Balme de Sillingy (74), Mulhouse (68), and Méry-sur-Oise (95). Fig 2

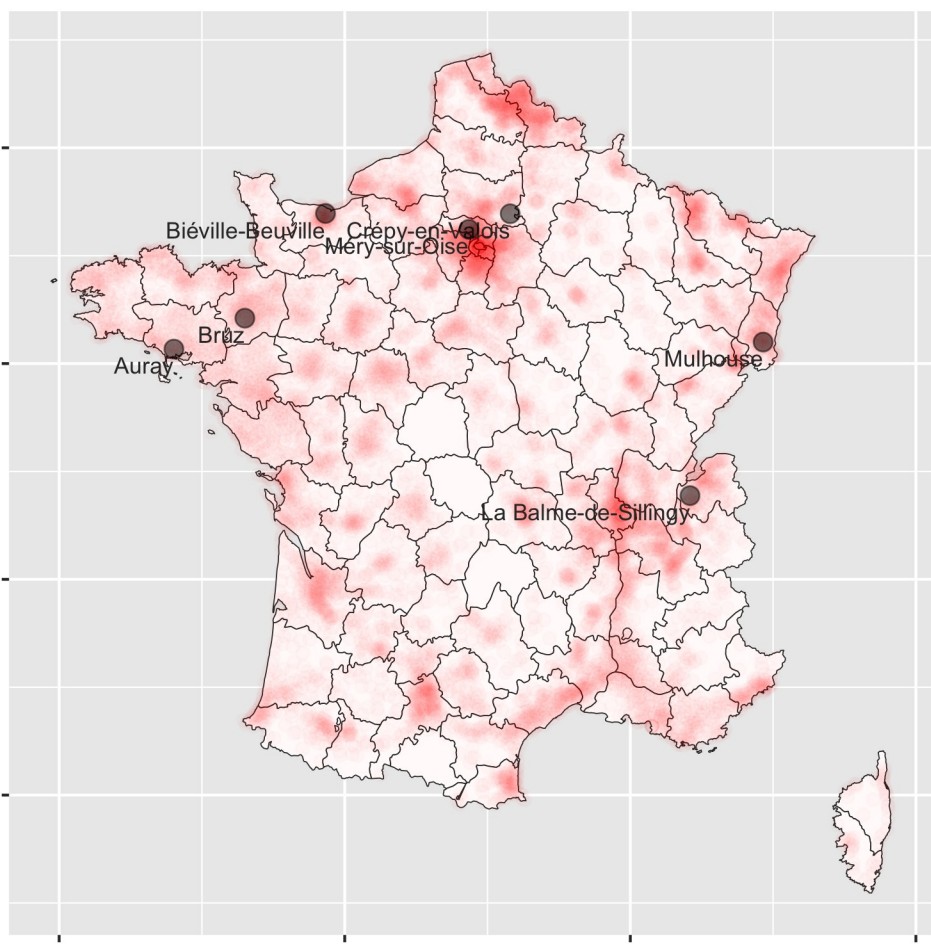

**Fig 2. Revealed COVID-19 clusters in time of election and number of cases released afterwards.** The map is created in R (3.5.3) [17] using package ggplot2 [18].

shows the location of the seven COVID-19 clusters according to the media at the time of election as well as the number of COVID-19 cases in France per *département*.

We then calculate the distance between each municipality and the nearest COVID-19 cluster and use it as a proxy of perceived exposure to the virus in the locality. In auxiliary robustness analysis, we compare the results we obtain using this proxy with those obtained with the official COVID-19 statistics such as hospitalization rates and deceased individuals. These alternative statistics are not our first choice for two main reasons: i) they were unknown at the time of the election but made available by public health authorities only after the ballot; ii) they are available at a more aggregate, departemental level (96 observations), unlike our distance measure that we compute at the municipality level (9,000 observations).

Second, we assume that the cost associated with COVID-19 is strongly related to voter age. It is reasonable to assume that the perceived risk of severe illness from COVID-19 increases with age. Because we do not have access to individual-level data, and we do not know the actual distribution of voters ages, we focus on the proportion of older voters per municipality. We therefore split the sample into two sub-samples, "Young" and "Old", according to the proportion of older inhabitants in the municipality. We defined a municipality to be "Young" ("Old") if the share of population aged 65 years or more is less (more) than the national

average of 19.22%. Note that the results are quite similar when adopting a 75-years-or-more threshold.

In addition, we consider population density as another proxy for the probability of becoming infected at the time of voting, regardless of the distance to the nearest cluster. We assume that more interactions (induced by higher density) increases the risk of interpersonal infection. Moreover, a higher density makes most voting stations more crowded.

Finally, we use the number of running lists to work out the probability of a voter being pivotal. More specifically, elections with a single running list have a very low stake, because there is no uncertainty about the winner. Elections with two running lists trigger more interest among voters as the final outcome of the election strongly depends on the results of the first round. This is all the more true when the number of running lists is greater than two (see S5 Section in S1 Appendix, for precise descriptions of data and variables).

## Empirical approach

We hypothesize that voters behave in a rational way and engage in cost-benefit analysis when they decide whether to vote or abstain. The cost of the COVID-19 pandemic is higher for older voters, and the perceived risk of being infected is higher in areas closer to the COVID-19 clusters known at election time and with high population density. We therefore hypothesize the following: municipalities with a i) larger proportion of older voters, ii) higher density and iii) closer to the seven officially identified COVID-19 clusters experience lower turnout rates. We also hypothesize that there is an interaction effect between the proportion of older voters and COVID-19 proxies. Our basic regression model reads as:

$$y_{ijt} = \beta + \theta post_t + \beta covid_i + \delta post_t \times covid_i + X'_{ijt}\eta + c_i + d_j + \epsilon_{ijt} \qquad (1)$$

where $y_{ijt}$ is the outcome variable, turnout rate, for municipality $i$, which belongs to *département j* at time $t$, i.e. 2014 and 2020. $post_t$ is an indicator variable that equals one for 2020 elections and zero for 2014 elections. $covid_i$ is a variable that measures the prevalence of the COVID-19 pandemic in municipality $i$ at the time of election. It is measured by a series of variables. Our key variable of interest is the distance, in kilometers, between municipality $i$ and the nearest cluster. We use it both as a discrete and continuous variable. Two other variables of interest in this context are the age structure of the voting-eligible population in the municipality, on the one hand, and the population density, on the other hand. They are part of $X_{ijt}$, a vector of variables that includes geographic variables such as a municipality's city size, type, and status (large/small, rural/urban); economic variables such as median income, unemployment, and share of homeowners; as well as socio-demographic variables such as education level, and number of registered voters. It also includes a set of political variables, such as the number of running lists, and the percentage of votes in favor of the leading party of the governmental coalition (LREM) at the previous presidential election (held in 2017). We controlled, in alternative models, for the potential influence of the vote for populist parties, and in particular that of Marine Le Pen (far-right) and Jean-Luc Mélenchon (far-left) at the same elections; results were quite similar. Given that this pandemic began in China and first arrived in Europe through northern Italy, it could be perceived as associated with population displacement. As a result, to capture the magnitude of contact with tourists and visitors among the local population, we include the number of hotel rooms *per capita*, in log scale. Finally, we include either municipality $c_i$ or *département $d_j$* fixed effects in some specifications, to control for unobserved heterogeneity.

Additionally, we run a set of first-difference regressions (see S6 Section in S1 Appendix) based on the difference in turnout rates between 2014 and 2020, to discard any unobserved

municipality-specific heterogeneity that might drive the results of our cross-sectional analysis (see Eq [1]). Here, we focus on the change in turnout rate over time. The results we obtain with this first-difference specification (see S7 Section in S1 Appendix) are quite similar to those obtained with a difference-in-differences approach, which we interpret in the next section.

Last, we estimate a triple-difference version of the model, along with a natural spline regression, to assess the potential non-linear effect of the distance to the nearest COVID-19 clusters on turnout rates to take into account the age structure of municipalities ("Old" versus "Young"). The model reads as follows:

$$y_{ijt} = \alpha_0 + \alpha_1 post_t + \alpha_2 covid_i + \alpha_3 old_i + \beta_1 post_t \times old_i + \beta_2 post_y \times covid_i +$$
$$\beta_3 old_i \times covid_i + \delta post_t \times covid_i \times old_i + X_{ijt}\eta + c_i + d_j + \epsilon_{ijt}$$

where $covid_i$, as above, is a variable that measures the prevalence of the COVID-19 pandemic in municipality $i$. Specifically, in some models $covid_i$ is just a dummy variable indicating whether a municipality is "Near" a COVID-19 cluster or "Far" from it (it takes on the value 1 if a municipality is "Near" COVID-19). The coefficient of interest in this version of the model, which is close to the double-difference approach adopted earlier on two separate samples ("Old" and "Young" municipalities), is $\delta$. The OLS estimate $\hat{\delta}$ is

$$\hat{\delta} = [(\bar{y}_{ON1} - \bar{y}_{ON0}) - (\bar{y}_{YN1} - \bar{y}_{YN0})] - [(\bar{y}_{OF1} - \bar{y}_{OF0}) - (\bar{y}_{YF1} - \bar{y}_{YF0})]$$

where the subscripts O, Y, F, and N denote "Old", "Young", "Far", and "Near" municipalities, respectively. The 0 and 1 subscripts indicate whether we are in 2020 (1) or 2014 (0). Thus, $\bar{y}_{ON1}$ is the average turnout in "Old" municipality, "Near" a COVID-19 cluster in the 2020 elections.

The OLS estimate $\hat{\delta}$ can also be written as

$$\hat{\delta} = (\Delta\bar{y}_{ON} - \Delta\bar{y}_{YN}) - (\Delta\bar{y}_{OF} - \Delta\bar{y}_{YF})$$

where $\Delta$ is the first difference over time of turnout for a given municipality.

## Main results

Fig 3 illustrates the difference-in-differences estimation results on three different sub-samples of municipalities located within 50km (sample 1: black dot), 50- 100km (sample 2: dark-gray dot) and beyond 100km (sample 3: light-gray dot) of the main clusters, respectively. This approach is equivalent to a triple-difference estimation in which the distance parameter is formally integrated in the model, as in the next section. We observe a significant and substantial drop in turnout, all other things being equal. This shows that, once we control for the potential influence of a set of confounding factors, compared to the 2014 election, turnout rates on average drop by about 20 percentage points in 2020. In line with our expectations, it also clearly highlights some differences, mainly between municipalities located close to COVID-19 clusters (sample 1) and those remotely located from those clusters (sample 3).

To interpret the interaction effects, Fig 4 illustrates the main estimation results for three different sub-samples of municipalities: those located within less than 50 km of the seven COVID-19 clusters (panel a), those located between 50 and 100 km of COVID-19 clusters (panel b), and those located beyond 100 km of COVID-19 clusters (panel c). It shows that in municipalities that are close to the COVID-19 clusters, turnout decreases with higher proportions of older voters (panel a). In contrast, in municipalities that are far away from the COVID-19 clusters, participation increases with proportions of older voters (panel c). When the covariates are fixed at means, we estimate turnout to be 52.99% in close municipalities

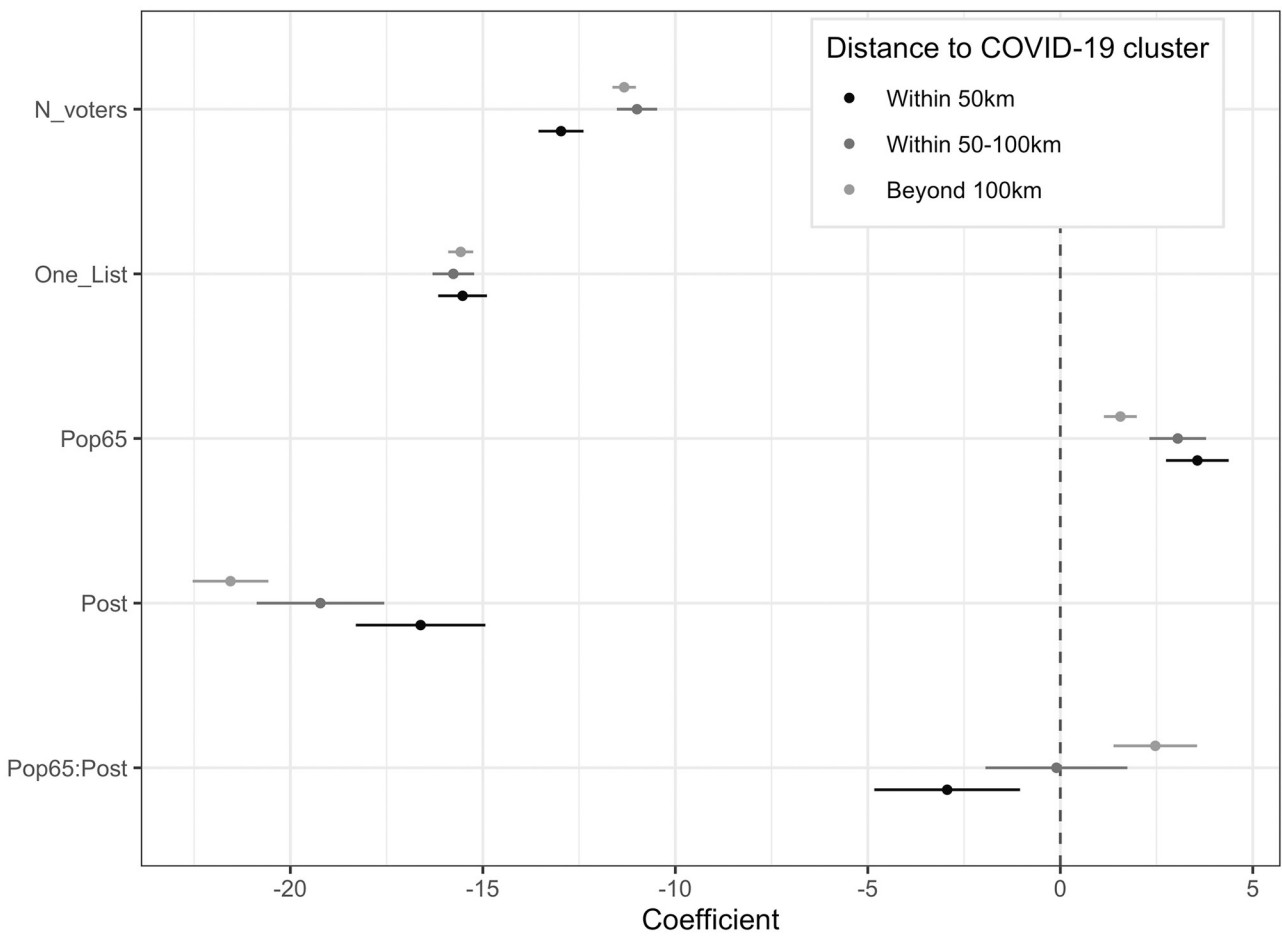

**Fig 3. Determinants of turnout (breakdown by distance to nearest COVID-19 clusters).**

and 57.99% in remote municipalities, leading to a mean difference of 4.98% between the two samples.

Alternatively, we split the sample into two different sub-samples, "Young" and "Old" municipalities (as defined before), and then interact the distance from COVID-19 clusters with our post dummy variable (*post × covid*). The coefficient of the interaction term is insignificant in the entire sample. However, it becomes positive and significant for municipalities with a higher proportion of older voters, and is negative and significant for municipalities with a lower proportion of older voters (see S5 Table in S1 Appendix).

Our estimates also include a series of controls, such as unemployment rate, annual income, share of votes for Emmanuel Macron at the 2017 presidential election, proportion of farmers, proportion of homeowners, incumbent mayor running for reelection, number of running lists, average education level, population, population density, number of hotel rooms *per capita*, and a region-fixed effect. Here we use an old map of French metropolitan regions in 22 different political entities. In alternative specifications, we used a finer set of 95 dummies for *département*s, which correspond to more decentralized political entities in France (findings are quite similar, as shown in S7 Section in S1 Appendix). Results are summarized in Fig 5. First of all, turnout rates are strongly and negatively affected by population density (in logs), which is in line with our expectations. In 2020 (respectively in 2014), a 10% increase in

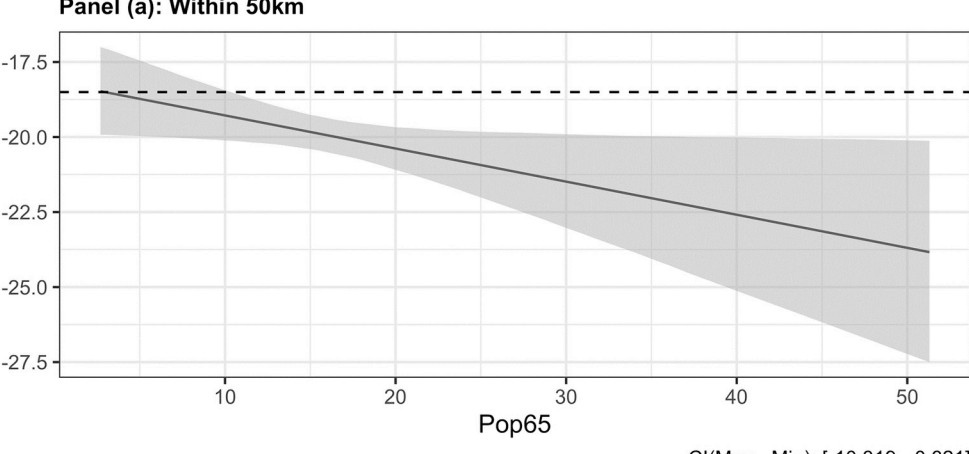

CI(Max - Min): [-10.319, -0.321]

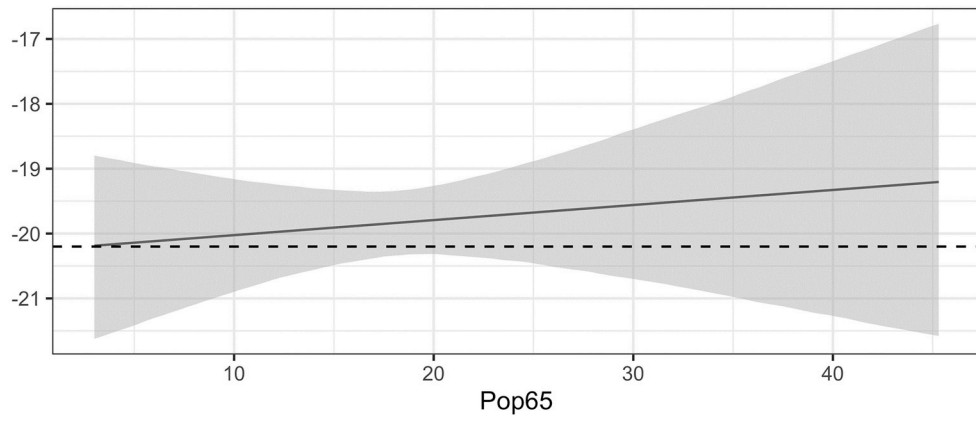

CI(Max - Min): [-2.612, 4.701]

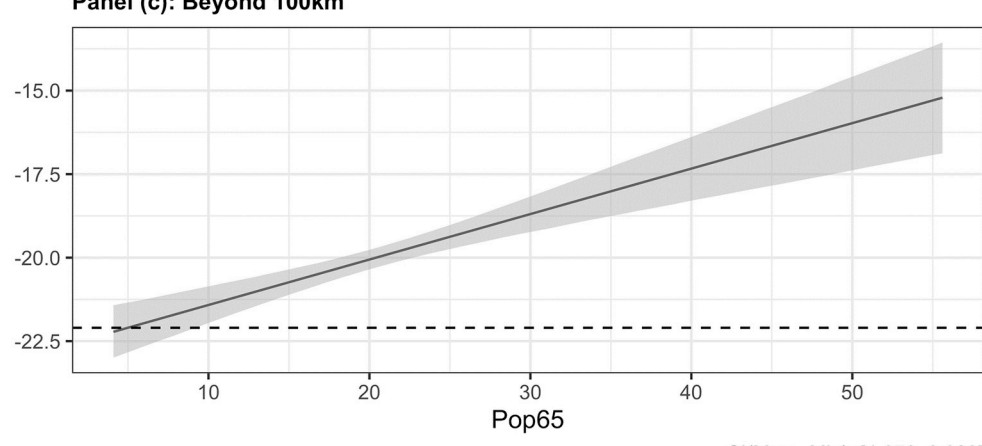

CI(Max - Min): [4.656, 9.328]

**Fig 4. Impact of the share of old voters on turnout (breakdown by distance to nearest COVID-19 clusters).** The vertical axis shows the percentage change (decrease) between 2014 and 2020. The models include *Unique list* and *Number of registered voters* as control variables.

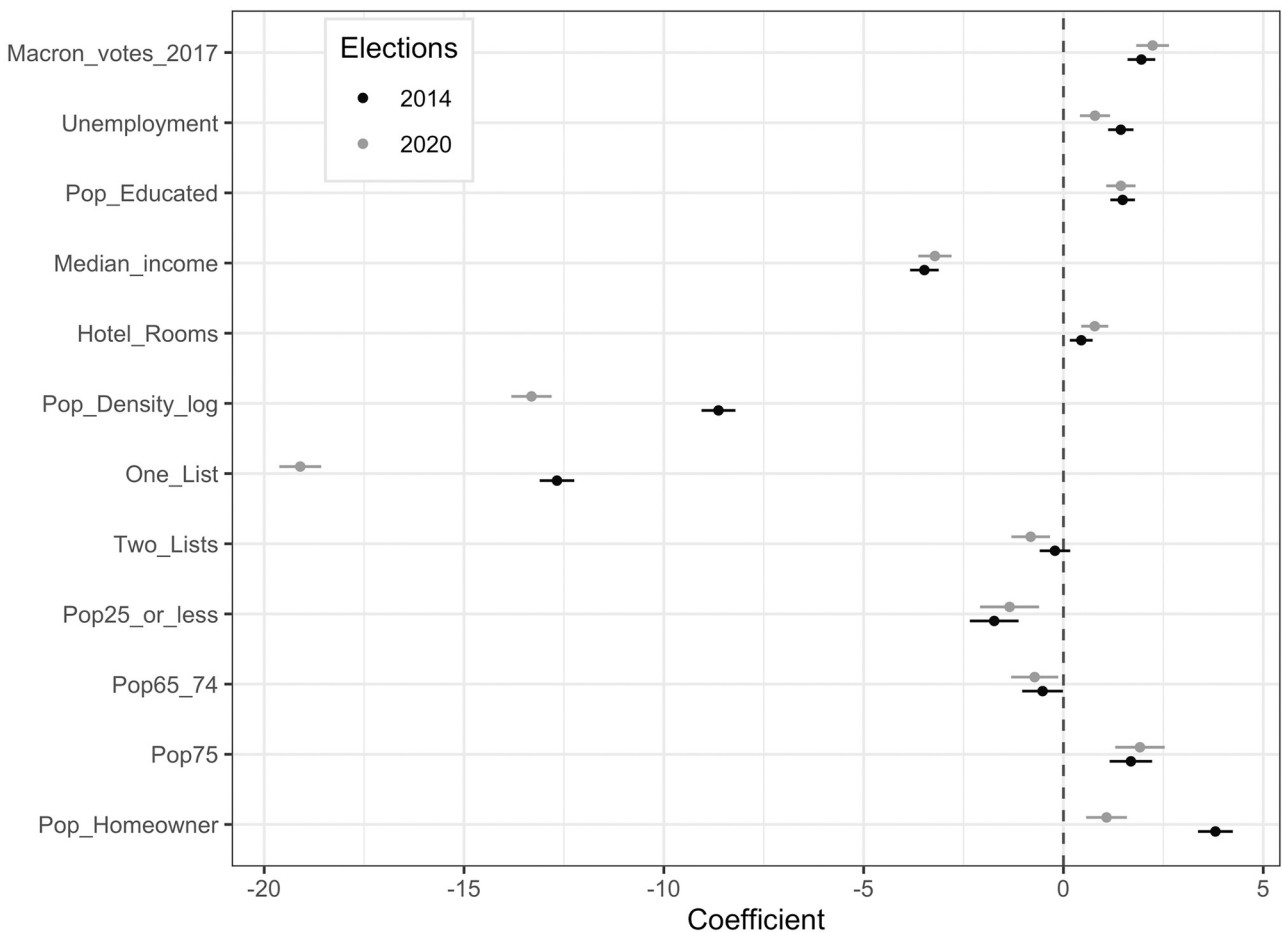

**Fig 5. Estimated coefficients at the 2014 and 2020 elections for turnout rate estimations.**

population density decreases turnout rates by 1.24% (respectively 0.76%). Interestingly, the impact of this variable is significantly larger in 2020, compared to 2014. This illustrates that the perceived risk of becoming infected when voting was higher in municipalities with more people per square kilometer around polling stations.

The impact of the number of running lists is large in magnitude in both elections. Overall, we observe a sharp difference between municipalities with a unique list (i.e., where the outcome of the poll was known in advance) and municipalities where the number of lists is larger than one (and the outcome of the vote was uncertain). Unsurprisingly, participation is lowest in municipalities where there is only one candidate running, because the probability of being the decisive voter is null, and the effect is amplified in the 2020 elections, compared to 2014. Relative to municipalities with unique lists, participation is, all other things being equal, 20 percentage points higher in municipalities where two candidates are running. The difference from the reference category (three candidates or more) is significant at around -1 percentage point in 2020. It is not significant in 2014. This set of results clearly shows that, even during a pandemic, the calculus of voting prevails.

Participation is higher in municipalities with a greater proportion of citizens aged 75 or older. This result is similar across the three local elections we have analyzed (2008, 2014 and 2020). It shows that the elderly consider local decisions particularly important to their

everyday life. For example, the proportion of homeowners within this category of voters is large and owners pay taxes determined at the local level. Mayors are also in charge of managing retirement homes, etc. In contrast, it decreases with the proportion of younger voters (aged 25–65) and slightly decreases with the proportion of the elderly aged 65–75. As expected, municipalities with a higher share of homeowners participate more. Turnout rates are higher on average in municipalities where the proportion of citizens who voted for Emmanuel Macron in 2017 at the last presidential election is higher, and where the incumbent mayor is running for reelection. The coefficients of economic variables such as median income and unemployment are also significant. The income coefficient is negative, while the unemployment coefficient is positive.

## Triple difference and spline regressions

To estimate the triple-difference model, also known as a Difference-in-difference-in-differences (DiDiD) model, we ran regressions of turnout using three dummy variables: "Post", "Near", and "Old". "Post" has already been defined in the previous section. "Near" takes on the value 1 if the municipality is located within 100km of a COVID-19 cluster, 0 otherwise. Finally, "Old" takes on the value 1 if the proportion of population aged 65+ is above the average (19.22%), and 0 otherwise. We estimate this model both in level and in first-difference. Our key variable of interest is the triple interaction "Post?Near?Old" when the model is estimated in level and "Near × Old" in first-difference models.

The results of our DiDiD models are reported in Table 1. Columns 1-4 are models in level, while columns 5-6 are first-difference models. Columns 1 and 4 do not include any additional control variables. Columns 2 and 5 include the set of control variables that we used in the previous section. In addition, Model (3) includes municipality fixed effects. The constant provides an estimate of the average turnout rate for the baseline or reference group. The reference group in columns 1-4 is 2014, "Young", and "Far" municipalities. For columns 4 and 5, the constant gives the average change (decrease here) over time in those municipalities. Overall,

**Table 1. Triple difference estimation results.**

| Variables | Level Analysis | | | First Difference Analysis | |
|---|---|---|---|---|---|
|  | (1) | (2) | (3) | (4) | (5) |
| Post | -21.019*** | -19.602*** | -19.737*** |  |  |
| Near | -4.306*** | -3.427*** |  | 1.097*** | 1.171*** |
| Old | 1.573*** | 0.750*** | -0.737*** | 1.351*** | 0.993*** |
| Post × Near | 0.994*** | 0.915*** | 1.123*** |  |  |
| Post × Old | 1.204*** | 1.469*** | 1.702*** |  |  |
| Near × Old | 1.544*** | 2.185*** | 0.647 | -1.737*** | -1.568*** |
| Post × Near × Old | -1.661*** | -1.842*** | -1.944*** |  |  |
| Constant | 67.470*** | 71.604*** | 79.169*** | -20.996*** | -20.201*** |
| Observations | 18,496 | 18,496 | 18,494 | 9,248 | 9,248 |
| R-squared | 0.534 | 0.814 | 0.940 | 0.003 | 0.701 |
| Control | No | Yes | Yes | No | Yes |
| Fixed Effects | No | No | Yes | No | No |

*** p<0.01,

** p<0.05,

* p<0.1

we observe as in the previous section, a sharp decline in participation rates from 2014 to 2020 by about 20 percentage points.

The estimated coefficient on those variables, which we call $\hat{\delta}$ is always negative and statistically significant at the 1% threshold, regardless of the specification we consider. On average, the impact of the proximity to COVID-19 clusters in 'Old' municipalities compared to 'Young' ones varies from -1.6 to -1.9 percentage points.

Finally, because the effect of proximity to the nearest COVID-19 cluster is likely to be non-linear, we also complement our analysis with natural spline regressions. Spline regressions, which allow us to account for non-linear effects, provide a more detailed analysis of proximity effects (see [19] for an application in a different setting).

Fig 6 summarizes the results of the natural spline regression approach in the triple-difference model. We used the ns package in R to generate a natural spline regression with three degrees of freedom (the results are similar when using a different number of degrees of freedom such as 4 or 5). We estimated a model in first difference, and include the set of control variables described above (we also estimated a model in level and obtained similar results, it is not reported here but is available upon request from the authors).

This figure illustrates the difference in turnout rates between 2020 and 2014 for "Old" and "Young" municipalities as a function of distance to COVID-19 clusters. The main noteworthy findings, are as follows. First, the change in turnout rates is negative for "Old" and "Young" municipalities. Second, compared to "Old" municipalities, there is a larger drop in turnout rate in "Young" municipalities, especially when considering the municipalities that are distant (beyond 80km) from COVID-19 clusters. For municipalities closer to COVID-19 clusters, i.e.

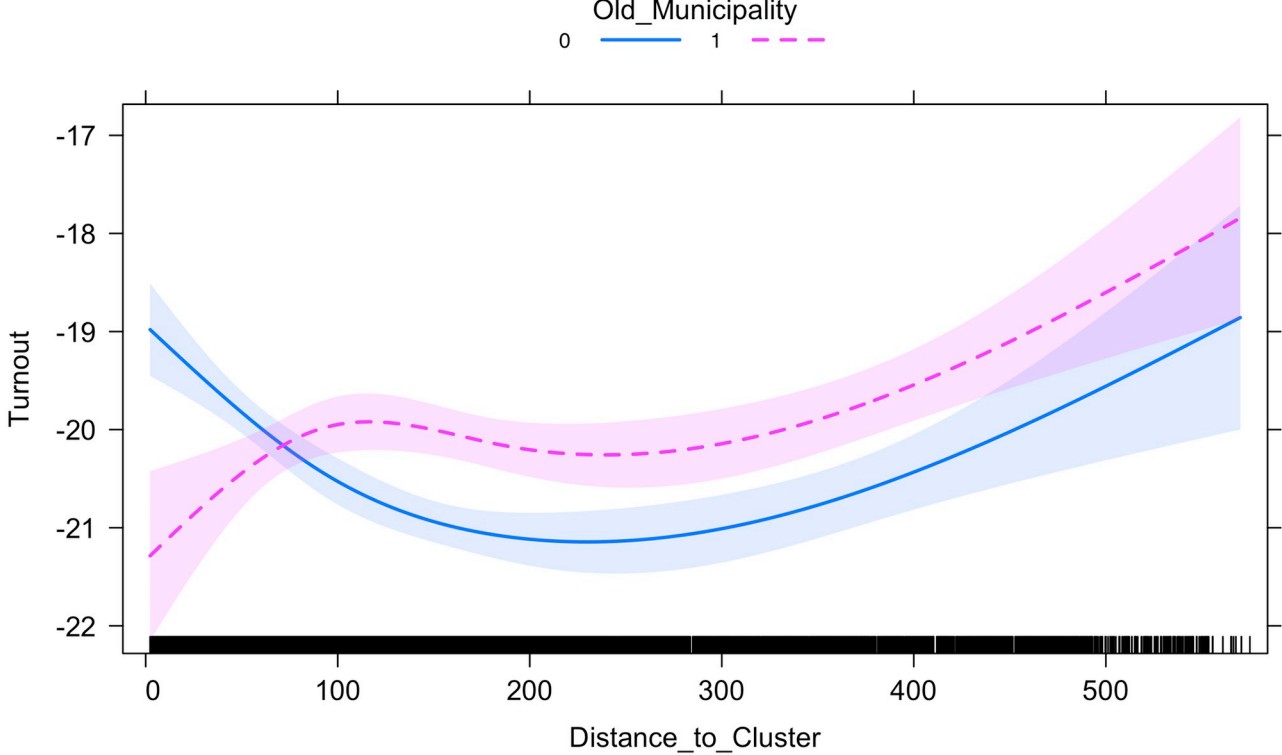

**Fig 6. Spline regression estimates.**

those located between 0 and 50km of clusters, the reverse is true. In "Old" municipalities, the closer a municipality is to a COVID-19 cluster, the larger the decline in turnout rate. Finally, beyond a distance of 430km, the significant difference between those two types of municipalities fades away.

Although the results presented in this section highlights a non-linearity, they are consistent with those presented in previous sections. The sanitary crisis had an effect in both groups, but they differ as a function of distance to COVID-19 clusters.

## Limitations

To make sure our estimates capture the effects of COVID-19-related factors, we considered a large number of confounding factors that vary across municipalities. The model with fixed effects, and the difference-in-differences strategy used in this paper, naturally removes the municipality-specific fixed effects, i.e. the unobserved drivers that do not vary over time. Nevertheless, our analysis is limited in a number of ways.

First, we do not have individual-level data, and only use aggregate data at the municipality level. One has to be aware of the so-called ecological fallacy, as in our empirical analysis we say nothing about individual behavior or decisions.

Second, we do not say much about the electoral winners and losers of COVID-19. For instance, one might argue that since COVID-19 directly stems from globalization, pro-globalization parties and lists, particularly those in close connection with parties in power, lose during a pandemic, while anti-globalization lists win in such a context. Although such questions are important, we are not able to answer them clearly, simply because we have no precise information about ideological views and opinions of lists in this local election. The media, focusing on a few high-profile cases in big cities, reported that the biggest winners of the delayed second round are the green party, Europe Ecologie Les Verts (EELV) (see for instance The Guardian, June 28, 2020).

Third, we cannot exclude the possibility that the sharp decline in turnout across municipalities is merely a continuation of a negative trend in French electoral participation. We also cannot exclude the role of time contextual effects. For instance, we cannot rule out that political dissent is an important driver of this low turnout rate, at a time of large protests in France. The yellow vest movement and the long December 2019 protest against ongoing pension reform initiated by the government are illustrations of the lack of support for the national government. We do not know to what extent these national factors play a role in the context of this election.

Finally, we have only focused on the first round of the municipal elections. The reason is that over 86% of municipalities did not have to organize a second round as a candidate was elected directly in the first round, gaining more than 50% of the votes. As a result, we end up with a very restricted sample of 1,330 municipalities in the second round. We note, however, that for these municipal elections the turnout remained low. Fig 7 illustrates the distribution of voter turnout change between the first and second rounds. It shows that there is little difference between 2014 and 2020, indicating that, as far as change in turnout is concerned, COVID-19 affected the first and second rounds of the elections similarly.

## Conclusions

The COVID-19 pandemic has the potential to have a massive negative impact on turnout rates, as we have shown occured in the 2020 municipal elections in France. Overall turnout decreased by a record 20 percentage points.

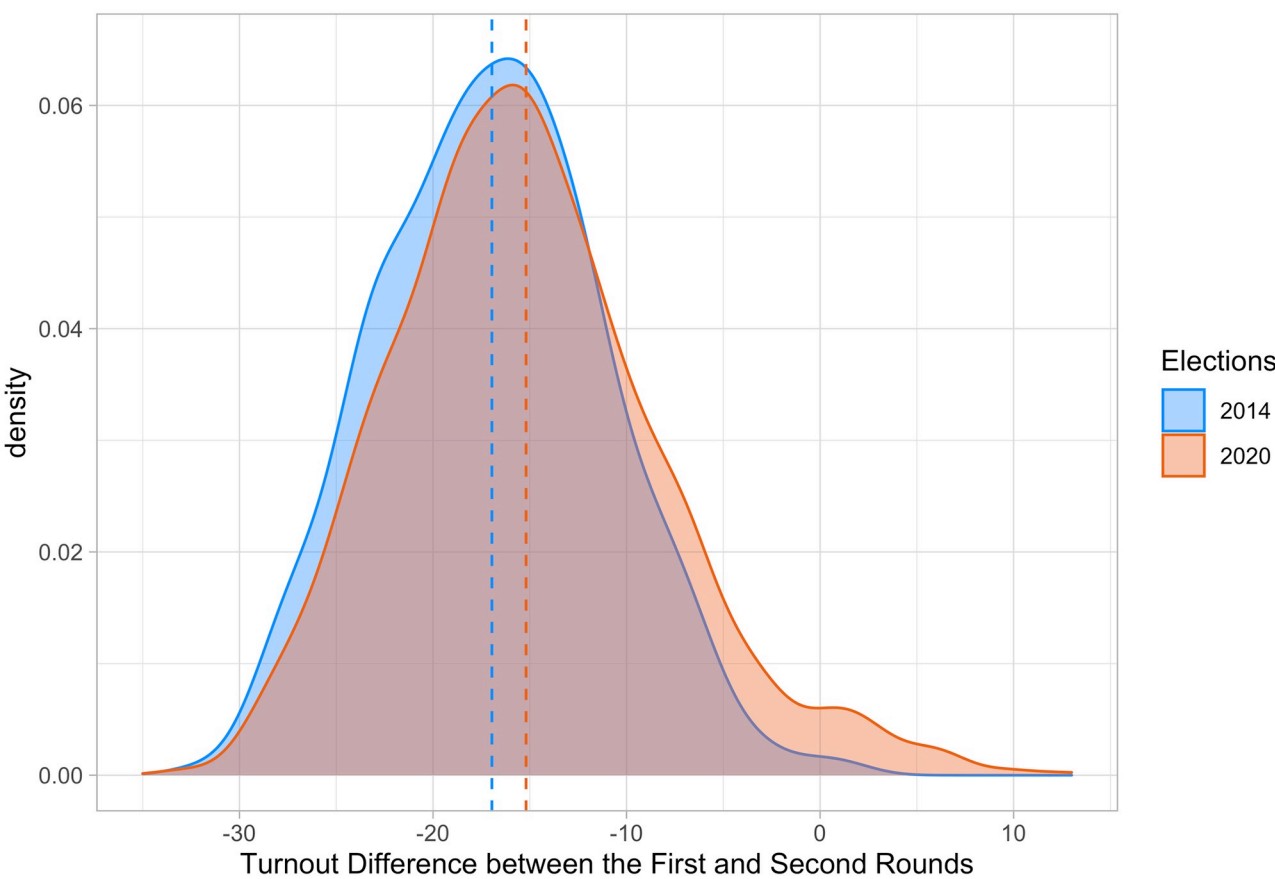

**Fig 7. Distribution of turnout rate changes between the first and second rounds in the 2014 and 2020 elections.**

We clearly show that COVID-19 negatively affected participation in municipalities with a larger fraction of elderly characterized by a higher risk to develop severe COVID-19 illness and in close proximity to COVID-19 clusters. Moreover, municipalities with a higher population density or where only one list was running at the election experienced a lower mobilization. The differentiated participation across population groups can have consequences for vote shares, as broadly discussed in political science [8, eg].

Although other factors also played a role, the COVID-19 pandemic was a strong driving force behind this historic fall. This result, obtained at local elections, should be confirmed in the context of national elections. However, the calculus of voting is identical at national elections as at local ones, so we expect a similar potential incidence of outbreak on national elections, all other things being equal. Among other things that vary from one election to another, the management of the election and the ballot change according to national practice and institutions. Since the election organization influences the cost of voting and the risk of being infected, both the extension of our results and the answer to the issues raised by unusually low participation rely on the solutions available to organize a ballot. The implementation of any voting methods reducing social interactions, and thus the cost of voting for citizens, would reduce the impact of outbreak on election outcomes. These well-known methods are early-voting, vote-by-mail, electronic voting, etc.

Beyond the debated question around the influence of low participation on election outcomes, this issue may have other major implications such as a lack of legitimacy of elected officials, which in turn may open the door to large dissent of the decisions they make.

## Supporting information

**S1 Appendix.**
(PDF)

## Acknowledgments

The authors would like to thank Quentin David, Christine Fauvelle-Aymar and LEM seminar participants at Lille University for their useful comments. All errors remain ours.

## Author Contributions

**Conceptualization:** Abel François.

**Data curation:** Alexandre Garel.

**Formal analysis:** Abdul Noury, Abel François, Olivier Gergaud, Alexandre Garel.

**Methodology:** Abdul Noury, Abel François, Olivier Gergaud.

**Supervision:** Abdul Noury, Olivier Gergaud.

**Writing – original draft:** Abdul Noury, Abel François, Olivier Gergaud.

**Writing – review & editing:** Abdul Noury.

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
