## [Decision Letter · Decision Letter 0]

16 Nov 2020

PONE-D-20-27759

How does COVID-19 affect Electoral Participation? Evidence from the French Municipal Elections

PLOS ONE

Dear Dr. Noury,

Thank you for submitting your manuscript to PLOS ONE. After careful consideration, we feel that it has merit but does not fully meet PLOS ONE’s publication criteria as it currently stands. Therefore, we invite you to submit a revised version of the manuscript that addresses the points raised during the review process. Upon reception of the revised manuscript from you, we will send it along to two reviewers who read your paper in the first round. 

We look forward to receiving your revised manuscript.

Kind regards,

Shang E. Ha, Ph.D.

Academic Editor

PLOS ONE

Journal Requirements:

3.We note that [Figure(s) 2] in your submission contain [map/satellite] images which may be copyrighted. All PLOS content is published under the Creative Commons Attribution License (CC BY 4.0), which means that the manuscript, images, and Supporting Information files will be freely available online, and any third party is permitted to access, download, copy, distribute, and use these materials in any way, even commercially, with proper attribution. For these reasons, we cannot publish previously copyrighted maps or satellite images created using proprietary data, such as Google software (Google Maps, Street View, and Earth). For more information, see our copyright guidelines: http://journals.plos.org/plosone/s/licenses-and-copyright.

1.    You may seek permission from the original copyright holder of Figure(s) [2] to publish the content specifically under the CC BY 4.0 license. 

Reviewers' comments:

Reviewer's Responses to Questions

**Comments to the Author**

1. Is the manuscript technically sound, and do the data support the conclusions?

Reviewer #1: Partly

Reviewer #2: Yes

2. Has the statistical analysis been performed appropriately and rigorously? 

Reviewer #1: No

Reviewer #2: Yes

3. Have the authors made all data underlying the findings in their manuscript fully available?

Reviewer #1: Yes

Reviewer #2: Yes

4. Is the manuscript presented in an intelligible fashion and written in standard English?

Reviewer #1: Yes

Reviewer #2: Yes

5. Review Comments to the Author

Reviewer #1: This article has an interesting starting point and research question, and the dataset employed is impressive. There are, however, a number of questions that arose while reading the manuscript.

The authors aim to test the effect of Covid19 on turnout, but in the process also bring in a host of other issues. Some of these have at best an indirect link to the Covid19 effect, such as the study of one-list municipalities or the population density effects (both of which also appear as ‘afterthoughts’ in the way they are brought forward in the manuscript). Others cause many more problems than they resolve, such as placing the 65-plus issue so central to the manuscript. The 65-plussers were not only urged to stay home (which reduces the authors’ ability to identify their ‘expected cost’ interpretation from the ‘stay home’ message), but also create a problem with aggregation bias (which undermines the authors’ ability to pinpoint the observed effects among the elderly). In a short piece of this kind, focus and credibility of identification is critical. Both are lacking to some extent in the presentation of the current manuscript.

That being said, I really liked the idea to exploit distance to known Covid19 hotspots as an identification strategy. This is very clever and useful. I think this should therefore be promoted to take the central focus of the manuscript. Using a DiD strategy is very clearly also helpful here (i.e. 2014 versus 2020 and close versus far), but more can be done to really exploit the distance effect. There is a large tradition to use natural spline regressions to address similar distance issues. I recently came across an articly applyingthis approach in a very different setting (Geys and Osterloh, 2013, Journal of Regional Science), but am convincced it will be useful here too. Moreover, when upgraded to be embedded within a DiD framework, this could really push the manuscript to the next level.

Note that the authors may then still introduce the elderly effect through a DiDiD approach. They in practice already appear to be heading towards this given the split-sample results including a postCovid-Pop65 interaction, but could take this idea more seriously and implement it more rigorously.

Also, the specification whereby infection cases at the time of election is used – which now suddenly appears without warning on p8 – could be upgraded to a proper robustness check.

All regressions should include municipality FEs to avoid interference from cross-sectional heterogeneity. The first-difference results are indeed comforting in that respect, but arguably would become superfluous once all main regressions include municipality FEs.

Finally, much of the conclusion is speculative and beyond the reach of the actual analysis. The conclusion should focus on what we can actually learn from the analysis.

Reviewer #2: I found this to be a very well-executed paper, and although in PLOS One this is not a criterion for assessing the merits of potential articles, it must be said that it focuses on one of the most timely topics in the field of electoral studies: what are the impacts of celebrating an election amid the pandemic? Methodologically, the paper is very sound, and the findings are clearly articulated and presented. It also dialogs with other working papers/current research related to this topic, which should be commended.

I would like to raise the following points:

#1 – The paper clearly makes the case that voter turnout declined especially in non-competitive races, which are taken to be those in which one list ran alone. But this looks like too crude an operationalisation of competitiveness. Including as an independent variable the margin of victory in the 2014 election, for instance, would allow for a more thorough assessment of the link between anticipated competitiveness and changes in voter turnout.

#2 – The reasons for including globalisation (measured by the number of hotel rooms per capita) as an independent variable are not clear enough; what exactly is the rationale for considering its role and what should we conclude from the coefficients?

#3 – The fact that the analysis is performed with data from what would be typically considered a second-order election (Reif and Schmitt 1980) could be the object of some deeper thoughts; to what extent do the authors consider that this circumstance influences the results and their potential of extrapolation to elections with a national realm?

6. PLOS authors have the option to publish the peer review history of their article (what does this mean?). If published, this will include your full peer review and any attached files.

Reviewer #1: No

Reviewer #2: No

---

## [Author Response · Author response to Decision Letter 0]

5 Jan 2021

“How does COVID-19 affect Electoral Participation? Evidence from the French Municipal Elections”

Abdul Noury, Abel Francois, Olivier Gergaud, and Alexandre Garel

Response to the Editor and the referees

Authors’ responses are in italics

We thank the referees for their interest in our paper and for their very helpful comments. We have addressed all of them in the revised version of the manuscript. We explain how in what follows.

Reviewer #1: 

This article has an interesting starting point and research question, and the dataset employed is impressive. There are, however, a number of questions that arose while reading the manuscript.

The authors aim to test the effect of Covid19 on turnout, but in the process also bring in a host of other issues. Some of these have at best an indirect link to the Covid19 effect, such as the study of one-list municipalities or the population density effects (both of which also appear as ‘afterthoughts’ in the way they are brought forward in the manuscript). Others cause many more problems than they resolve, such as placing the 65-plus issue so central to the manuscript. The 65-plussers were not only urged to stay home (which reduces the authors’ ability to identify their ‘expected cost’ interpretation from the ‘stay home’ message), but also create a problem with aggregation bias (which undermines the authors’ ability to pinpoint the observed effects among the elderly). In a short piece of this kind, focus and credibility of identification is critical. Both are lacking to some extent in the presentation of the current manuscript.

Answer:

We have revised the text to make clearer the usage we make of the variables. We hope it is now easier to understand.

That being said, I really liked the idea to exploit distance to known Covid19 hotspots as an identification strategy. This is very clever and useful. I think this should therefore be promoted to take the central focus of the manuscript. 

Answer:

Thanks for this suggestion. The main identification strategy we use is indeed to exploit the distance to known Covid-19 hotspots. However, based on the entire sample, in a difference-in-differences specification the interaction term between distance and post is not significant. Only when we split the sample between ‘Old’ and ‘Young’ municipalities, does the distance interaction term becomes significant, in a theoretically predictable way. As the referee suggested a DiD framework with split samples is equivalent to a DiDiD, which is now formally introduced in the text. 

Using a DiD strategy is very clearly also helpful here (i.e. 2014 versus 2020 and close versus far), but more can be done to really exploit the distance effect. There is a large tradition to use natural spline regressions to address similar distance issues. I recently came across an articly applying this approach in a very different setting (Geys and Osterloh, 2013, Journal of Regional Science), but am convinced it will be useful here too. Moreover, when upgraded to be embedded within a DiD framework, this could really push the manuscript to the next level.

Note that the authors may then still introduce the elderly effect through a DiDiD approach. They in practice already appear to be heading towards this given the split-sample results including a postCovid-Pop65 interaction, but could take this idea more seriously and implement it more rigorously.

Answer:

Thank you for this suggestion. We agree with the referee that this approach is the best strategy in the context of these elections and the main characteristic of the COVID-19 pandemic, which causes more mortality among the elderly population. Using a natural spline coupled with a DiDiD framework results in very insightful results. Indeed, Fig 6 shows that the impact of distance to the main COVID-19 clusters is different from one election to the other and across groups. 

Also, the specification whereby infection cases at the time of election is used – which now suddenly appears without warning on p8 – could be upgraded to a proper robustness check.

Answer:

Thank you for pointing out this issue. We have modified the text accordingly.

All regressions should include municipality FEs to avoid interference from cross-sectional heterogeneity. The first-difference results are indeed comforting in that respect, but arguably would become superfluous once all main regressions include municipality FEs.

Answer:

Thank you for this suggestion. We agree that including fixed effects are important to avoid interference from cross-sectional heterogeneity. The appendix of the paper contains a series of first-difference regressions, which removes the fixed effects. Following the referee’s suggestion, we explicitly include fixed effects in the DiDiD model. Our key variable of interest remains significant when we include municipality fixed effects.

Finally, much of the conclusion is speculative and beyond the reach of the actual analysis. The conclusion should focus on what we can actually learn from the analysis.

Answer:

Thank you for this suggestion. We have revised the text accordingly, especially the introduction and conclusion.

Reviewer #2: 

I found this to be a very well-executed paper, and although in PLOS One this is not a criterion for assessing the merits of potential articles, it must be said that it focuses on one of the most timely topics in the field of electoral studies: what are the impacts of celebrating an election amid the pandemic? Methodologically, the paper is very sound, and the findings are clearly articulated and presented. It also dialogs with other working papers/current research related to this topic, which should be commended.

I would like to raise the following points:

#1 – The paper clearly makes the case that voter turnout declined especially in non-competitive races, which are taken to be those in which one list ran alone. But this looks like too crude an operationalization of competitiveness. Including as an independent variable the margin of victory in the 2014 election, for instance, would allow for a more thorough assessment of the link between anticipated competitiveness and changes in voter turnout.

Answer:

We agree with the referee that a more refined measure of competitiveness would be desirable. 

However, the operationalization of competitiveness by the number of lists is the best way because of the effective number of lists. Indeed, the number of running lists is extremely low as described in the table below.

Elections 1 list 2 lists 3 lists 4 and more

2014 30.4% 42.5% 16.3% 10.8%

2020 37.6% 39.4% 13.3% 9.7%

In 2020, for more than a third of the municipalities studied in our paper, there is a unique list, meaning that the competitiveness measure for these municipalities, for instance the margin of victory, is null. So, the variable measuring the competitiveness of the election contains two things: the fact that there is more than one list and the competitiveness itself. Moreover, if you note that the municipalities with 2 lists represent almost 40% of the sample in 2020, the competitiveness is highly linked to the number of lists. That is why we decided to use the number of lists as proxy of the competitiveness, rather than a measure based on the votes’ dispersion. 

To follow the referee’s suggestion, we computed the margin of victory for 2014, and included it in our model (results available upon request). Our key results remained unchanged when included this variable. One practical difficulty in computing the margin of victory is that for municipalities with unique list there is no competition, and therefore the margin of victory is 100%. Including the margin of victory will result in dropping about 4000 observations due to perfect multicollinearity with our one_list variable. Consequently, we prefer to work with the number of lists as a measure of electoral competitiveness rather than margin of victory in 2014.

#2 – The reasons for including globalization (measured by the number of hotel rooms per capita) as an independent variable are not clear enough; what exactly is the rationale for considering its role and what should we conclude from the coefficients?

Answer:

This is a good point, thank you. We agree with the referee that the interpretation of this variable was somehow misleading in the first version of the text. We propose a different explanation in the current version and now consider this variable as a proxy used to capture the intensity with which the city is exposed to external visitors and therefore an increased risk of contamination coming from this non-local population.

#3 – The fact that the analysis is performed with data from what would be typically considered a second-order election (Reif and Schmitt 1980) could be the object of some deeper thoughts; to what extent do the authors consider that this circumstance influences the results and their potential of extrapolation to elections with a national realm?

Answer:

It is a particularly good point. We have added two discussions about this question in the new version of the text. First, we describe the importance of the municipal election within French politics as compared to other elections. So, we can not consider municipal elections as precisely second-order elections. Even though these are mid-term elections compared to the presidential election which in France is the first-order election, the local dimension, with strong local issues, is important. Moreover, the fragmentation of the French territory (there are more than 36,600 municipalities) makes very hard the emergence of national issues. Second, we introduce a discussion in the concluding section, which deals with the local characteristics of this election and the extension of our results to national elections.

---

## [Decision Letter · Decision Letter 1]

1 Feb 2021

How does COVID-19 affect Electoral Participation? Evidence from the French Municipal Elections

PONE-D-20-27759R1

Dear Dr. Noury,

We’re pleased to inform you that your manuscript has been judged scientifically suitable for publication and will be formally accepted for publication once it meets all outstanding technical requirements.

Kind regards,

Shang E. Ha, Ph.D.

Academic Editor

PLOS ONE

Additional Editor Comments (optional):

Reviewers' comments:

Reviewer's Responses to Questions

**Comments to the Author**

1. If the authors have adequately addressed your comments raised in a previous round of review and you feel that this manuscript is now acceptable for publication, you may indicate that here to bypass the “Comments to the Author” section, enter your conflict of interest statement in the “Confidential to Editor” section, and submit your "Accept" recommendation.

Reviewer #1: All comments have been addressed

Reviewer #2: All comments have been addressed

2. Is the manuscript technically sound, and do the data support the conclusions?

Reviewer #1: Yes

Reviewer #2: Yes

3. Has the statistical analysis been performed appropriately and rigorously? 

Reviewer #1: Yes

Reviewer #2: Yes

4. Have the authors made all data underlying the findings in their manuscript fully available?

Reviewer #1: Yes

Reviewer #2: Yes

5. Is the manuscript presented in an intelligible fashion and written in standard English?

Reviewer #1: Yes

Reviewer #2: Yes

6. Review Comments to the Author

Reviewer #1: (No Response)

Reviewer #2: I believe the manuscript addresses all the issues that I raised in the initial review round and think that it is now suited for publication.

7. PLOS authors have the option to publish the peer review history of their article (what does this mean?). If published, this will include your full peer review and any attached files.

Reviewer #1: No

Reviewer #2: No

---

## [Editor Report · Acceptance letter]

17 Feb 2021

PONE-D-20-27759R1 

How does COVID-19 affect Electoral Participation? Evidence from the 2020 French Municipal Elections 

Dear Dr. Noury:

I'm pleased to inform you that your manuscript has been deemed suitable for publication in PLOS ONE. Congratulations! Your manuscript is now with our production department. 

Kind regards, 

on behalf of

Dr. Shang E. Ha 

Academic Editor

PLOS ONE